# Health Professionals’ Counseling about Electronic Cigarettes for Smokers and Vapers in a Country That Bans the Sales and Marketing of Electronic Cigarettes

**DOI:** 10.3390/ijerph17020442

**Published:** 2020-01-09

**Authors:** Katia Gallegos-Carrillo, Inti Barrientos-Gutiérrez, Edna Arillo-Santillán, Luis Zavala-Arciniega, Yoo Jin Cho, James F. Thrasher

**Affiliations:** 1Epidemiology and Health Services Research Unit, Mexican Institute of Social Security, Cuernavaca, Morelos 62000, Mexico; 2Tobacco Research Department, National Institute of Public Health, Cuernavaca, Morelos 62100, Mexico; edna@insp.mx; 3Evaluation and Surveys Research Center, National Institute of Public Health, Cuernavaca, Morelos 62100, Mexico; inti.barrientos@insp.mx; 4School of Demography, The Australian National University, Canberra, Australia/Tobacco Research Department, National Institute of Public Health, Cuernavaca, Morelos 62100, Mexico; 5Department of Epidemiology, University of Michigan, Ann Arbor, MI 48109, USA; luiszavala171409@gmail.com; 6Department of Health Promotion, Education & Behavior, Arnold School of Public Health, University of South Carolina, Columbia, SC 29208, USA; ycho@email.sc.edu

**Keywords:** e-cigarettes, smoking cessation, public health, primary health care

## Abstract

This study describes the prevalence and correlates of adult smokers’ discussions about electronic cigarettes (e-cigarettes) with health professionals (HPs), including whether these discussions may lead smokers and vapers to use e-cigarettes for smoking cessation. Methods: We analyzed data from an online survey of Mexican smokers recruited from a consumer panel for marketing research. Participants who had visited an HP in the prior four months (*n* = 1073) were asked about discussions of e-cigarettes during that visit and whether this led them to try to quit. Logistic models regressed these variables on socio-demographics and tobacco use-related variables. Results: Smokers who also used e-cigarettes (i.e., dual users) were more likely than exclusive smokers to have discussed e-cigarettes with their HP (adjusted odds ratio (AOR) = 3.96; 95% C.I. 2.73, 5.74), as were those who had recently attempted to quit smoking (AOR = 1.89; 95% C.I. 1.33, 2.7). Of smokers who had discussed e-cigarettes, 53.3% reported that the discussion led them to use e-cigarettes in their quit attempt. Also, dual users (AOR = 2.6; 95% C.I. 1.5, 4.5) and daily smokers (>5 cigarettes per day) (AOR = 3.62; 95% C.I. 1.9, 6.8) were more likely to report being led by their HP to use e-cigarettes in the quit attempt compared to exclusive smokers and non-daily smokers, respectively. Conclusions: Discussions between HP and smokers about e-cigarettes were relatively common in Mexico, where e-cigarettes are banned. These discussions appear driven by the use of e-cigarettes, as well as by greater smoking frequency and intentions to quit smoking.

## 1. Introduction

The long-term health consequences of using electronic cigarettes (e-cigarettes), as well as their utility for smoking cessation [1,2,3,4], are uncertain. These uncertainties have made physician guidelines around e-cigarettes challenging, even though e-cigarettes are the most popular method smokers choose for quitting in high-income countries [5,6,7]. Studies in high-income countries have found that discussions about e-cigarettes between health professionals (HPs) and smokers are uncommon [8], regardless of the e-cigarette’s regulatory framework in those countries [9,10]. Furthermore, studies from the smoker’s point of view are scarce, refs. [8,9,10,11] with no known studies on this topic in low- and middle-income countries (LMICs) in spite of the increasing use of e-cigarettes in LMICs, including in LMICs that ban them, likely due to their relatively large informal economies [12,13]. Furthermore, as countries increasingly restrict and even ban e-cigarettes [13], there is a need to understand the frequency, content, and consequences of HPs discussions about e-cigarettes with their patients who smoke, in a country like Mexico where e-cigarettes are banned but yet widely accessible. 

Previous cross-sectional studies in high-income countries have found that current and higher frequency use of e-cigarettes are associated with discussing e-cigarettes and receiving counseling from an HP to use them to quit smoking [9,10]. E-cigarette discussions were also more prevalent among smokers who were male [9], younger, and with higher educational attainment [10]. Furthermore, in a nationally representative survey in the United States, discussions about e-cigarettes with their HP (i.e., physicians and dentists), were less likely among ex-users of e-cigarettes than among current e-cigarettes users, whether they exclusively used e-cigarettes or also smoked cigarettes [8]. Additionally, making quit attempts were associated with having talked with an HP about e-cigarettes [9], although the temporality and direction of association were not clear. 

Patient interest in having discussions about e-cigarettes with their HP appears high. One study reported that 24% of adult patients attending a family medicine clinic would want to have these discussions with their HPs; among recent e-cigarettes users, this proportion was double (62%) [14]. To our knowledge, there are no studies from LMICs on HP discussions about e-cigarettes with their patients.

Mexico is a middle-income country where the importation, distribution, marketing, and sales of e-cigarettes are banned, as in most other Latin American countries [15] and, increasingly, around the world [16]. Nevertheless, in 2016, 12% of middle schoolers had vaped in the past 30 days [17] and 18% of Mexican adult smokers had tried e-cigarettes [18]. A longitudinal study concluded that Mexican smokers who used e-cigarettes were no more likely than exclusive smokers to quit smoking or reduce cigarette consumption [19], which is consistent with some studies in other countries where device type and relatively frequent e-cigarettes use were not considered [20,21]. Furthermore, in 2009, only 17%–19% of Mexican smokers who had visited an HP in the last year received smoking cessation counseling from their HP, which is much lower than other Latin American countries like Brazil (57%) or Argentina (60%) [22,23].

How HPs in Mexico or other LMICs approach e-cigarettes are not known. As e-cigarettes use rapidly increases worldwide, it is necessary to examine HP counseling practices that integrate Nicotine Vaping Products (NVP) information, whether for discouraging or for promoting e-cigarettes use for smoking cessation and harm reduction or for discouraging their use. In the end, HP discussions and their effects may be critical to advancing public health goals to reduce the toll of tobacco product use, particularly as the science and optimal regulations for e-cigarettes and other novel nicotine products evolve. Our study aims to describe the prevalence and correlates of adult smokers’ discussions about e-cigarettes with their HPs in Mexico, including whether these discussions may lead smokers to use e-cigarettes for smoking cessation. 

## 2. Materials and Methods

### 2.1. Design and Study Population

This study included Mexican adult smokers aged 18 or higher and who smoked in the last 30 days. In each of three separate survey waves (24 November–10 December 2018; 16 March–8 April 2019; 17 July–9 August 2019), 1500 participants were recruited through an online commercial panel for marketing research, using quotas for education (i.e., at least 500 with high school or lower attainment) and current e-cigarette use (at least 500) at each survey wave. While some participants were followed from one wave to the next, the present study includes data only from the first survey to which participants responded (wave 1, *n* = 1501; wave 2, *n* = 1035; wave 3, *n* = 799), limiting the analytic sample to those who indicated that they had consulted with an HP in the 4 months prior to the survey (*n* = 1073), which represents the number of participants in this study. Surveys took between 20 and 25 minutes to complete, on average, and all study procedures were approved by the Institutional Review Board and Ethics Committee of the National Institute of Public Health of Mexico. Participants reported socio-demographic information, as well as smoking- and NVP-related perceptions and behaviors. 

### 2.2. Measures

#### 2.2.1. Outcome Variables

##### Health Professional Consultation

Participants were asked questions adapted from prior research on HP discussions about e-cigarettes [10], in which participants were first asked if, during the prior 4 months (the period between surveys), they had a medical consultation with an HP, including a general practitioner, nurse or other health professionals (yes vs. no). Those who reported doing so were asked: “On any visit to a doctor or health professional in the last 4 months, did a health professional talk to you about e-cigarettes?” (yes; no; don’t know). Those who responded “yes” were asked: 1. “The last time you discussed e-cigarettes with a doctor or health professional, did you bring it up or did they?”(the doctor or health professional brought it up = 1; I brought it up = 0; don’t know = 0); and 2. “What advice did the doctor or health professional give you about e-cigarettes?” (they specifically recommended that I use e-cigarettes = 1; they advised me against using e-cigarettes = 0; they didn’t express a view for or against e-cigarette use = 0). Finally, participants were asked, “Did the conversation with your doctor or health professional lead you to make a quit attempt?” (Yes, and I used an e-cigarette in that quit attempt = 1; Yes, but I didn’t use the electronic cigarette in that attempt = 0; the discussion did not persuade me to try to quit smoking = 0).

#### 2.2.2. Independent Variables

##### Smoking- and E-Cigarettes-Related Variables

Participant responses to questions on combustible cigarette use in the last 30 days were used to determine the frequency of use: (a) non-daily smoker; (b) daily smoker, ≤5 cigarettes per day; and (c) daily smoker, >5 cigarettes per day. Among daily smokers, five cigarettes per day is the median cut point in prior research with Mexican smokers [24], including in representative samples [25], as in the sample for this study. Those who indicated that they had used e-cigarettes in the last 30 days were classified as dual users. Other smoking-related variables included having attempted to quit smoking in the prior 4 months (yes vs. no) and intentions to quit smoking within the next six months (yes vs. no). In addition, participants were asked if during a medical consultation in the last 4 months, an HP had counseled them to quit smoking (yes vs. no). 

##### Demographics Variables

Participants reported their sex (male or female), age (i.e., 18–29, 30–39, 40–49, or 50+ years), highest educational attainment (less than high school, high school graduate or technical studies or some college, college degree or postgraduate studies), and monthly household income in Mexican pesos (i.e., <$8000; $8001–$15,000; $15,001–$20,000; >$20,000), where the exchange rate was approximately $20 pesos to $1 US dollar. 

### 2.3. Statistical Analyses

Descriptive analyses were conducted to determine the frequency of electronic cigarette discussion according to socio-demographic and tobacco use variables. 

Bivariate and multivariate logistic regression models were estimated to determine the socio-demographic and smoking-related correlates of discussing e-cigarette use among smokers who reported an HP consultation. Furthermore, in the group who discussed e-cigarettes, logistic regression models estimated the socio-demographic and tobacco-use related correlates of the HP bringing up the topic, the HP recommending e-cigarettes use, and if discussions with the HP led them to use e-cigarettes to quit smoking. All statistical models were adjusted by socio-demographic (sex, age, educational attainment, and monthly household income) and smoking- and e-cigarette-related variables (frequency of use, any quit attempts in the prior 4 months, and intentions to quit smoking within the next six months). Analyses were conducted using STATA 15.1 (StataCorp 2017, College Station, TX, USA).

## 3. Results

Of survey participants who were eligible for the study, all who had consulted an HP in the prior 4 months comprised the analytic sample (*n* = 1073; see Table 1). The mean age of respondents was 36.6 years, about half (52.4%) were male, and almost 40% had high school education or lower. Of those who consulted an HP in the prior 4 months, 41% were dual users, and 49.2% had attempted to quit attempt during that same period of time.

### 3.1. HP Discussions about E-Cigarettes 

Among smokers and vapers who had an HP consultation during the last 4 months, 33.7% (*n* = 362) discussed e-cigarettes with their HP (See Table 2). Dual users were significantly more likely to discuss with their HP about e-cigarettes (adjusted odds ratio (AOR) = 5.1; 95% C.I. 3.7, 7.2) than exclusive combustible cigarette smokers (see Table 2). Respondents who had an attempt to quit smoking in the last 4 months were also more likely to discuss e-cigarettes (AOR = 1.9; 95% C.I. 1.4, 2.7). In addition, those who reported their HP counseling them to quit smoking during the consultation were more likely to discuss e-cigarettes (AOR = 3.34; 95% C.I. 2.4, 4.6); after controlling for demographic and smoking-related variables, *p*-values < 0.05 were considered statistically significant.

### 3.2. Content of E-Cigarettes Discussions with an HP and Advice about Smoking Cessation 

Among those who discussed e-cigarettes with their HP (*n* = 362), 46% reported that the HP brought up the topic. In adjusted models, the only statistically significant correlate was being a dual user (AOR = 1.74; 95% C.I. 1.05, 2.9; see Table 3). 

Almost half (46%) of smokers who discussed e-cigarettes indicated that their HP recommended their use, 23.5% reported being advised against their use and 29.6% indicated their HP did not express an opinion either for or against e-cigarettes use (Table 4). Furthermore, smokers who reported that their HP had counseled them to quit smoking were more likely to report that their HP recommended them to use e-cigarettes (AOR 1.7; 95% C.I. 1.0, 2.7).

### 3.3. Use of E-Cigarettes in Quit Attempts Following Health Professional Discussions

Among respondents who talked about e-cigarettes with their health professional, 53.3% reported that the discussion led them to use an e-cigarettes in a subsequent quit attempt, though 32.4% reported that the discussion persuaded them to have a quit attempt without using e-cigarettes in that attempt, while 14.3% indicated that their discussion carried out with the HP did not lead them to make a quit attempt. (Table 5). Dual users were more likely than exclusive smokers to be persuaded to use e-cigarettes in their quit attempt (AOR = 2.6; 95% C.I. 1.5, 4.5). Being a daily smoker relative to nondaily smoker was also significantly associated with using e-cigarettes in their quit attempt, with a weaker association for those who smoked less than 5 cigarettes per day (AOR = 1; 95% C.I. 1.0, 3.5) than for those who smoked 5 cigarettes per day or more (AOR = 3.6; 95% C.I. 1.9, 6.8). Additionally, respondents reporting that they plan to quit during the next six months (AOR = 1.7; 95% C.I. 1, 2.8) and whose HP advised them to quit smoking at all were more likely to be persuaded by their HP to use e-cigarettes in their quit attempt (AOR = 2.0; 95% C.I. 1.2, 3.4). 

## 4. Discussion

Health professionals’ counseling on smoking cessation is a fundamental component of tobacco control, with evidence from recent trials [26,27] leading some agencies to recommend that all physicians discuss e-cigarettes with their patients who smoke [28], while others suggest the evidence is not sufficient for physician recommendations [29]. Nevertheless, our study suggests that a nontrivial percentage of HPs discussing e-cigarettes with their patients who smoke (34%), even in Mexico, where e-cigarettes are banned. Studies in other countries have found a lower prevalence, ranging from 4% in Australia to 15% in studies in the U.S. [8,9,12,30]. The composition of our sample, which purposefully includes a higher percentage of e-cigarettes users, likely explains the higher prevalence we found. Furthermore, our relatively recent data collection effort may reflect the growing use of NVPs in Mexico. 

Studies in the U.S. exploring this topic from the physician point of view indicated a higher frequency of these discussions among patients who smoke, the prevalence varies from 48% [31] to 65% [32], and 70% [33], in studies carried out in the U.S. This gap in the perspectives of health professionals and patients about e-cigarettes discussions during health consultations indicates the importance of future research to clarify what accounts for these differences in perceptions.

Consistent with prior research [9], we found that being a dual user and having recently attempted to quit smoking were associated with NVP discussions. As with those other studies, the temporal sequence of HP consultation, e-cigarette use, and quit attempts are not clear. However, our analysis indicated that a significant percentage of smokers who discussed their e-cigarette use with their HP reported going on to try to quit with an e-cigarette. Nevertheless, they were still smoking at the time of the interview; hence, the quit attempt was not successful. Longitudinal research is sorely needed to help determine whether HPs can successfully promote quitting by encouraging e-cigarette use, including whether additional behavioral support is needed to achieve this goal, as suggested by recent clinical trials [24]. In the end, the greatest public health benefits would come from getting smokers who use e-cigarettes to quit e-cigarettes as well. However, previous studies in Mexico have found that only 3% of adult smokers who have tried to quit report that they received smoking cessation counseling from an HP [34]—a substantial amount of work remains to be done in this area [13,30,35,36,37]. 

It is noteworthy that participants reported that HPs initiated discussions about e-cigarettes, despite the fact that they are banned in Mexico. However, e-cigarettes are widely available, perhaps because of the size of the informal economy, which represents 56.9% of all jobs in Mexico [38]. Furthermore, e-cigarette purchase and consumption are not illegal. Hence, it is likely that most Mexicans do not know about the e-cigarette ban, including physicians. Future research should explore physician perspectives about opportunities to promote e-cigarettes for the cessation and harm reduction in the context of bans, especially given recent concerns about vaping-related illnesses and their apparent link to black-market products, particularly cannabis oils. 

Limitations of this study include our use of a convenience sample recruited from a consumer marketing panel. As the sample came from an unknown sampling frame, the generalizability of results is not clear; however, the sample was purposefully recruited to represent key market segments in Mexico, even though it over-represented higher socioeconomic status groups. Furthermore, because of our desire to have statistical power for evaluating the impact of e-cigarette use, we oversampled smokers who also used e-cigarettes, which likely led to overestimates of e-cigarette discussions and their use in quit attempts. Furthermore, our study did not disentangle issues around the temporal ordering of events and potential recall bias related to counseling and content of HP discussions from the participants’ points of view. However, our questions asked about a shorter time frame (i.e., the prior 4 months) than in other studies (e.g., last year [8,10]), which likely reduced recall bias. Finally, the few online survey questions we used to characterize the content of conversations with HPs may not capture key dynamics of the interaction (e.g., trust or competing clinical demands) that matter for the cessation of smoking. Future qualitative research on this topic should be considered to more comprehensively characterize and address the complexity and associated effects of these discussions. 

## 5. Conclusions

Discussions and recommendations to use e-cigarettes by physicians appear prevalent in Mexico, particularly amongst smokers who also use e-cigarettes. When the discussion of electronic cigarettes (e-cigarettes) took place, almost half of HPs brought up the topic and recommended their use, while more than half of smokers reported going on to use electronic cigarettes (e-cigarettes) in a quit attempt. This suggests that HPs may have an important role to play in smoking cessation that involves e-cigarettes. Future studies are needed to assess smoking cessation outcomes that follow from the HP encounter, including clinical trials to determine best clinical practices for promoting smoking cessation, including e-cigarette use for harm reduction, as well as the cessation of all nicotine products.

## Figures and Tables

**Table 1 ijerph-17-00442-t001:** Sample characteristics of smokers and vapers having health professional (HP) consultations during the prior 4 months, aged 18 to 71, and living in Mexico 2018–2019.

Variables	*n* = 1073
	(%)
**Age group**	
18–29	33.2
30–39	31.4
40–49	16.3
50+	16.3
**Sex**	
Female	52.4
Male	47.6
**Education**	
Less than high school	7.2
High School graduate	33.2
Some college	19.8
College degree or higher	39.9
**Household income**	
≤8000	21.8
8000–15,000	28.2
15,000–20,000	17.4
>20,000	28.3
Missing	4.2
**Type of user**	
Exclusive Cigarette user	59
Dual user	41
**Smoking frequency and intensity**	
Non-daily	53.8
Daily ≤ 5 cigs	20.5
Daily > 5 cigs	25.7
**Quit attempt (last 4 months)**	
No	50.8
Yes	49.2
**Plan to quit**	
I have no plans/>6 months/future	56.9
During the next month/1–6 months	43.1

**Table 2 ijerph-17-00442-t002:** Characteristics associated with having a discussion with their HP about electronic cigarettes among adult smokers aged 18 to 71, living in Mexico 2018–2019, (*n* = 1073).

	Discussion with Health Professionals about E-Cigarettes				
	*n* (%)	Unadjusted Model		Adjusted Model ^a^	
**Variables**	362 (33.7)	O.R.	CI 95%	A.O.R.	CI 95%
**Age**					
18–29	40 **	*Reference*		*Reference*	
30–39	37	0.99	0.73, 1.34	1.02	0.69, 1.5
40–49	23	0.47	0.31, 0.71 **	0.64	0.38, 1.08
50 and more	22	0.41	0.27, 0.61 **	0.83	0.49, 1.39
**Sex**					
Female	29 **	*Reference*		*Reference*	
Male	40	1.61	1.25, 2.1 **	1.32	0.95, 1.82
**Education**					
Less than high school graduate	24 **	Reference		Reference	
High school graduate or technical	28	1.24	0.69, 2.21	1.36	0.67, 2.7
Some college	30	1.32	0.72, 2.42	1.21	0.6, 2.6
College Degree or Higher	43	2.49	1.42, 4.37 *	1.69	0.8, 3.5
**Income**					
≤8000	39	Reference		Reference	
8000–15,000	32	0.71	0.50, 1.03	0.6	0.38, 0.96 *
15,000–20,000	31	0.71	0.47, 1.08	0.44	0.26, 0.76 *
>20,000	36	0.94	0.66, 1.34	0.55	0.33, 0.93 *
missing	17	0.29	0.12, 0.68	0.57	0.2, 1.6
**Type of user ***					
Exclusive Cigarette user	16 **	Reference		Reference	
Dual user	58	7.04	5.26, 9.43 **	5.14	3.65, 7.2 **
**Smoking frequency and intensity**					
Non-daily	31	Reference		Reference	
Daily ≤ 5 cigs	34	1.12	0.8, 1.57	1.33	0.9, 2
Daily > 5 cigs	38	1.30	0.96, 1.77	1.44	0.96, 2.1
**Quit attempt (last 4 months)**					
No	20 **	Reference		*Reference*	
Yes	47	3.57	2.71, 4.7 **	1.89	1.33, 2.7 **
**Plan to quit**					
I have no plans/>6 months/future	23 **	Reference		Reference	
During the next month/1–6 months	47	2.87	2.2, 3.75 **	1.68	1.19, 2.4 *
**During medical consultation in the last 4 months had**					
**counseling to quit smoking**					
No	19 **	Reference		Reference	
Yes	54	4.91	3.74, 6.46 **	3.34	2.4, 4.6 **

* *p* < 0.05, ** *p* < 0.01. O.R. Odds Ratio, A.O.R. Adjusted Odds Ratio; ^a^ Adjusted model included all variables specified in the table (147 missing data in family income variable).

**Table 3 ijerph-17-00442-t003:** Characteristics associated with the physician bringing up electronic cigarettes with adult smokers aged 18 to 71, living in Mexico 2018–2019 (*n* = 366).

	Who Brought the Discussion about E-Cigarettes (Physician vs. Patient)				
	Physician (*n* %)	Unadjusted Model		Adjusted Model ^a^	
**Variables**	168 (45.9)	**O.R.**	C.I. 95%	**A.O.R.**	C.I. 95%
**Age**					
18–29	41	Reference		Reference	
30–39	50	0.66	0.41, 1.06	0.64	0.38, 1.1
40–49	42	0.83	0.41, 1.68	0.90	0.4, 2.1
50 and more	57	0.49	0.25, 0.98 *	0.53	0.24, 1.2
**Sex**					
Female	45	Reference		Reference	
Male	48	0.82	0.54, 1.25	0.77	0.48, 1.2
**Education**					
Less than high school graduate	44	Reference		Reference	
High school graduate or technical	56	0.63	0.23, 1.75	0.45	0.14, 1.45
Some college	40	1.21	0.42, 3.5	0.85	0.24, 2.9
College Degree or Higher	44	1.05	0.39, 2.79	0.75	0.23, 2.5
**Income**					
≤8000	49	Reference		Reference	
8000–15,000	47	1.15	0.64, 2.1	0.94	0.48, 1.8
15000–20,000	39	1.68	0.85, 3.29	1.49	0.67, 3.3
>20,000	47	1.12	0.64, 1.96	1.03	0.49, 2.2
missing	43	1.33	0.28, 6.3	0.86	0.15, 4.8
**Type of user ***					
Exclusive Cigarette user	56	Reference		Reference	
Dual user	43	1.79	1.12, 2.87 *	1.74	1.05, 2.9 *
**Smoking frequency and intensity**					
Non-daily	48	Reference		Reference	
Daily ≤ 5 cigs	47	1	0.58, 1.74	0.94	0.52, 1.7
Daily > 5 cigs	45	1.16	0.71, 1.91	1.24	0.71, 2.2
**Quit attempt (last 4 months)**					
No	49	Reference		Reference	
Yes	46	1.11	0.7, 1.75	0.89	0.53, 1.5
**Plan to quit**					
I have no plans/>6 months/future	49	Reference		Reference	
During the next month/1–6 months	46	1.16	0.75, 1.79	1.14	0.7, 1.9
**During medical consultation in the last 4 months had counseling to quit smoking**					
No	49	Reference		Reference	
Yes	45	1.21	0.78, 1.88	1.2	0.72, 1.9

* *p* < 0.05, ** *p* < 0.01. O.R. Odds Ratio, A.O.R. Adjusted Odds Ratio; ^a^ Adjusted model included all variables specified in the table (147 missing data in family income variable).

**Table 4 ijerph-17-00442-t004:** Factors associated with physicians recommending e-cigarettes to adult smokers (among those who discussed e-cigarettes) aged 18 to 71, living in Mexico 2018–2019, (*n* = 375).

	Health Professional Recommendation: They Specifically Recommended Use E-Cigarettes				
	*n* (%)	Unadjusted Model		Adjusted Model ^a^	
**Variables**	*n* = 171 (46%)	O.R.	C.I. 95%	A.O.R.	CI 95%
**Age**					
18–29	42	Reference		Reference	
30–39	54	1.57	0.98, 2.51	1.6	0.94, 2.7
40–49	44	1.1	0.55, 2.2	1.05	0.5, 2.3
50 and more	39	0.85	0.44, 1.66	0.8	0.4, 1.7
**Sex**					
Female	43	Reference		Reference	
Male	49	1.23	0.82, 1.85	1.11	1.1, 0.69, 1.7
**Education**					
Less than high school graduate	50	Reference		Reference	
High school graduate or technical	45	0.83	0.3, 2.28	0.74	0.23, 2.4
Some college	50	1.00	0.35, 2.84	0.84	0.24, 2.9
College Degree or Higher	45	0.77	0.29, 2.04	0.61	0.18, 2.01
**Income**					
≤8000	38	Reference		Reference	
8000–15,000	57	2.11	1.18, 3.8 *	2.62	1.3, 5.1 *
15,000–20,000	50	1.78	0.91, 3.47	2.14	0.97, 4.7
>20,000	47	1.25	0.71, 2.2	1.57	0.74, 3.3
missing	3	1.29	0.27, 6.14	1.83	0.32, 10.2
**Type of user**					
Exclusive Cigarette user	48	Reference		Reference	
Dual user	47	0.97	0.61, 1.54	0.91	0.55, 1.5
**Smoking frequency and intensity**					
Non-daily	44	Reference		Reference	
Daily ≤ 5 cigs	45	1.06	0.6, 1.85	0.95	0.52, 1.7
Daily > 5 cigs	53	1.43	0.87, 2.36	1.3	0.75, 2.2
**Quit attempt (last 4 months)**					
No	44	Reference		Reference	
Yes	48	1.19	0.75, 1.88	1.06	0.63, 1.8
**Plan to quit**					
I have no plans/>6 months/future	41	Reference		Reference	
During the next month/1–6 months	51	1.48	0.96, 2.29	1.5	0.93, 2.5
**During medical consultation in the last 4 months had**					
**counseling to quit smoking**					
No	38 *	Reference		Reference	
Yes	50	1.65	1.05, 2.59 *	1.66	1.03, 2.7 *

* *p* < 0.05, ** *p* < 0.01. O.R. Odds Ratio, A.O.R. Adjusted Odds Ratio; ^a^ Adjusted model included all variables specified in the table (147 missing data in family income variable). Reference: They did not express a view for or against e-cigarettes/do not know/advice against the use of e-cigarettes.

**Table 5 ijerph-17-00442-t005:** Factors associated with adult smokers using e-cigarettes to quit smoking because of HP consultation, adults aged 18 to 71, living in Mexico 2018–2019, (*n* = 362).

	Health Professional Persuade You to Use E-Cigarettes to Quit Smoking ^a^				
	*n* (%)	Unadjusted Model		Adjusted Model ^b^	
**Variables**	*n* = 193 (53.3)	O.R.	C.I. 95%	A.O.R.	C.I. 95%
**Age**					
18–29	57	Reference		Reference	
30–39	53	0.84	0.52, 1.36	0.76	0.43, 1.34
40–49	45	0.91	0.45, 1.82	0.84	0.25, 2
50 and more	40.9	0.52	0.26, 1.03	0.4	0.17, 0.92 *
**Sex**					
Female	57	Reference		Reference	
Male	50	0.75	0.49, 1.14	0.71	0.43, 1.19
**Education**					
Less than high school graduate	39	Reference		Reference	
High school graduate or technical	53	1.77	0.63, 4.96	1.45	0.41, 5.1
Some college	66	2.99	1.01, 8.8 *	2.68	0.68, 10.5
College Degree or Higher	51	1.62	0.6, 4.4	0.92	0.25, 3.4
**Income**					
≤8000	42	Reference		Reference	
8000–15,000	64	2.41	1.33, 4.4 *	1.64	0.81, 3.3
15,000–20,000	55	1.68	0.86, 3.27	1.24	0.54, 2.8
>20,000	54	1.6	0.91, 2.8	1.39	0.63, 3.1
missing	29	0.54	1, 2.97	0.72	0.1, 4.8
**Type of user ***					
Exclusive Cigarette user	38 **	Reference		Reference	
Dual user	60	2.44	1.5, 3.4 **	2.6	1.5, 4.5 **
**Smoking frequency and intensity**					
Non-daily	44 *	Reference		Reference	
Daily ≤ 5 cigs	57	1.66	0.95, 2.9	1.88	1.01, 3.5 *
Daily > 5 cigs	68	2.63	1.6, 4.4 **	3.62	1.9, 6.8 **
**Quit attempt (last 4 months)**					
No	49	Reference		Reference	
Yes	56	1.33	0.84, 2.1	0.93	0.54, 1.6
**Plan to quit**					
I have no plans/>6 months/future	47	Reference		Reference	
During the next month/1–6 months	58	1.51	0.97, 2.33	1.68	1.0, 2.8 *
**During medical consultation in the last 4 months had counseling to quit smoking**					
No	45 **	Reference		Reference	
Yes	57	1.65	1.05, 2.6 *	2	1.2, 3.4 *

* *p* < 0.05, ** *p* < 0.01. O.R. Odds Ratio, A.O.R. Adjusted Odds Ratio. ^a^ Sample size includes only respondents who discussed e-cigarettes with their HP. ^b^ Adjusted model included all variables specified in the table (147 missing data in family income variable). Reference: The discussion did not persuade me to try to quit smoking/Yes but I did not use the electronic cigarette in that attempt.

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
