# Peer review of "Health Professionals’ Counseling about Electronic Cigarettes for Smokers and Vapers in a Country That Bans the Sales and Marketing of Electronic Cigarettes"

_ijerph, 2020, doi:10.3390/ijerph17020442_

Round 1
Reviewer 1 Report
• I would like to commend the authors on a well-written paper that makes an important contribute to our understanding of factors related to NVP use--in this case, in a middle-income country. My feedback (detailed below) pertains mostly to the methods used.
• This paper uses “NVP” throughout the manuscript, but if I'm not mistaken, participants were recruited based on e-cigarette use and the survey questions asked only about e-cigarettes. As such, I found it a bit misleading for the manuscript to use the term NVP--at least in the methods, results, and abstract. It is worth explicitly noting to readers that this investigation focuses on one type of NVP, e-cigarettes, but that that this work has implications for NVP research more broadly.
• Methods: Given that NVPs are banned in Mexico, do the researchers have any reason to suspect that participants were not honest in reporting their NVP use or their NVP-related discussions with clinicians? Given that ~1/3 reported discussing NVPs with a provider, I doubt this was the case. Nonetheless, from the perspective of self-report data, it is a question worth asking when studying the use of illegal substances. If anything, I wonder if the data reported here underestimate the actual prevalence of NVP discussions.
• Methods: Interpersonal conversations are complex interactions that will require more nuanced quantitative (and qualitative) data than this investigation can offer. For example, other variables likely play a role in predicting whether smokers and dual users are persuaded by their clinicians to make a quit attempt, such as trust in the provider, demographic concordance between the provider and the patient (e.g., gender, age), valence of the conversation, etc. To be sure, the current data are a productive starting point, but I would encourage the authors to note these matters in the limitations section.
• Methods: The phrasing of the survey question "Did a health professional talk to you about e-cigarettes?" gives me pause. Because "health professional" is the subject of the sentence and "you" is the direct object, participants could have inferred that this question was specifically asking about provider-initiated conversations. Even though the following question asked participants about who initiated the conversation, if the initiation question appeared on a separate page from the first question, participants may have inferred that the first question was asking exclusively about conversations that the provider initiated. So if a participant interpreted the first question as asking about provider-initiated conversations but they (the patient) were the one who initiated the conversation, the participant would indicate "no." As a consequence, the rate of people reporting an NVP conversation (33.7%) would be overestimated. A more neutral phrasing would have been something like, "On any visit to a doctor or health professional in the last 4 months, did e-cigarettes come up at all in conversation?”
• Methods: To split daily smokers into heavy and light daily smokers, what was the rationale for using 5 cigarettes as the cutoff?
• Methods: Do the researchers have data on which NVPs participants are using aside from e-cigarettes (if any)? If so, that would be useful descriptive information to include in Table 1.
• Discussion: To me, the most compelling finding is that about half of the conversations reported by participants were initiated by providers…in a country where NVPs are banned. Yet this was not mentioned in the discussion. I would encourage the authors to weave this into the discussion if possible. It begs the question of whether providers’ opinions regarding NVPs (or at least, e-cigarettes) are not aligned with government policy.
• Discussion, line 187: Who's to say this rate is high in absolute terms? Relative to estimates from other countries, the rate is relatively higher, sure. Maybe "nontrivial" would be better.
Author Response
General comment: I would like to comment the authors on a well-written paper that makes an important contribute to our understanding of factors related to NVP use--in this case, in a middle-income country. My feedback (detailed below) pertains mostly to the methods used. |
Answer: We would like to thank the Reviewer for giving us this feedback which allowed us to improve this manuscript.
|
Comment: This paper uses “NVP” throughout the manuscript, but if I'm not mistaken, participants were recruited based on e-cigarette use and the survey questions asked only about e-cigarettes. As such, I found it a bit misleading for the manuscript to use the term NVP--at least in the methods, results, and abstract. It is worth explicitly noting to readers that this investigation focuses on one type of NVP, e-cigarettes, but that that this work has implications for NVP research more broadly. |
Answer: We appreciate this comment and now use the more comment term "electronic cigarettes" throughout this manuscript.
|
Comment: Methods: Given that NVPs are banned in Mexico, do the researchers have any reason to suspect that participants were not honest in reporting their NVP use or their NVP-related discussions with clinicians? Given that ~1/3 reported discussing NVPs with a provider, I doubt this was the case. Nonetheless, from the perspective of self-report data, it is a question worth asking when studying the use of illegal substances. If anything, I wonder if the data reported here underestimate the actual prevalence of NVP discussions. |
Answer: As we mention in our manuscript (see page 2, lines 69-71), while the importation, distribution, advertising and sales of e-cigarettes are banned, this law is not enforced and e-cigarettes appear widely available in both the formal and informal economies. Also, the purchase and consumption of e-cigarettes is not illegal, which likely explains the high level of reported discussions with providers in our study. Indeed, e-cigarette use is even more common than smoking among middle schoolers. As such, we believe that the vast majority of Mexicans do not know about the illegality of e-cigarette sales. We bring up this issue in the limitations section of the discussion, where we also raise the reviewers concerns (see page 11, lines 226-232).
|
Comment: Methods: Interpersonal conversations are complex interactions that will require more nuanced quantitative (and qualitative) data than this investigation can offer. For example, other variables likely play a role in predicting whether smokers and dual users are persuaded by their clinicians to make a quit attempt, such as trust in the provider, demographic concordance between the provider and the patient (e.g., gender, age), valence of the conversation, etc. To be sure, the current data are a productive starting point, but I would encourage the authors to note these matters in the limitations section. |
Answer: We appreciate this comment and now address it as a limitation of our study that merits further research (page 11, lines 242-246) |
Comment: The phrasing of the survey question "Did a health professional talk to you about e-cigarettes?" gives me pause. Because "health professional" is the subject of the sentence and "you" is the direct object, participants could have inferred that this question was specifically asking about provider-initiated conversations. Even though the following question asked participants about who initiated the conversation, if the initiation question appeared on a separate page from the first question, participants may have inferred that the first question was asking exclusively about conversations that the provider initiated. So if a participant interpreted the first question as asking about provider-initiated conversations but they (the patient) were the one who initiated the conversation, the participant would indicate "no." As a consequence, the rate of people reporting an NVP conversation (33.7%) would be overestimated. A more neutral phrasing would have been something like, "On any visit to a doctor or health professional in the last 4 months, did e-cigarettes come up at all in conversation?” |
Answer: We agree with the reviewer that this is a potential issue, but that the results (i.e., high prevalence of discussions) suggest that it was not. The Spanish language version of this sentence may not be limited in the way that the reviewer suggests. Hence, we only indicate the source of the questions that we adapted for our study (page 3, lines 106-107).
|
Comment: Methods: To split daily smokers into heavy and light daily smokers, what was the rationale for using 5 cigarettes as the cutoff? |
Answer: We now include our rationale for using a 5 cigarette/day cutoff point for daily smokers, which is based on the median cigarettes per day for daily smokers in our sample and in larger, nationally representative surveys (GATS Encuesta Global de Tabaquismo en Adultos, México, 2009). Indeed, the frequency of smoking among Mexicans and US Latinos of Mexican ancestry is generally much lower than in other ethnic groups (Kaplan et al., 2014; Zhu, Pulvers, Zhuang, & BaezcondeGarbanati, 2007), as we now indicate in the methods section, page 3, lines 124-126. · GATS Encuesta Global de Tabaquismo en Adultos, México. (2009). Mexico: Ministry of Health. · Kaplan, R. C., Bangdiwala, S. I., Barnhart, J. M., Castañeda, S. F., Gellman, M. D., Lee, D. J., et al. (2014). Smoking among U.S. Hispanic/Latino adults: The Hispanic community health study/study of Latinos. American Journal of Preventive Medicine, 46(5), 496–506. · Zhu, S. H., Pulvers, K., Zhuang, Y., & Baezconde-Garbanati, L. (2007). Most Latino smokers in California are low-frequency smokers. Addiction, 102(Suppl. 2), 104–111. |
Comment: Methods: Do the researchers have data on which NVPs participants are using aside from e-cigarettes (if any)? If so, that would be useful descriptive information to include in Table 1. |
Answer: Mexicans in our sample only use e-cigarettes, as heated tobacco products were banned and generally unavailable during the study period. However, in the period since data collection, transnational tobacco companies have started to market these products and we will collect data on their uptake and impact on other tobacco product use. We hope that our changing the terminology throughout the paper to “e-cigarettes” addresses this issue. |
Comment: Discussion: To me, the most compelling finding is that about half of the conversations reported by participants were initiated by providers…in a country where NVPs are banned. Yet this was not mentioned in the discussion. I would encourage the authors to weave this into the discussion if possible. It begs the question of whether providers’ opinions regarding NVPs (or at least, e-cigarettes) are not aligned with government policy. |
Answer: We have added a paragraph to the discussion to address this issue (page 11, lines 244-247) |
Comment: Discussion, line 187: Who's to say this rate is high in absolute terms? Relative to estimates from other countries, the rate is relatively higher, sure. Maybe "nontrivial" would be better |
Answer: We have followed the reviewer’s recommendation to change “high” for “nontrivial” (page 10, line 202).
|

Reviewer 2 Report
Dear Editor,
Thank you for giving me the opportunity to review the manuscript entitled “Health professionals’ counseling about e-cigarettes for smokers and vapers in a country that bans electronic cigarette sales and marketing”.
The manuscript covers very important and actual topic and it is in the scope the Journal.
Please find below several comments which in my opinion can improve the paper:
The title would be slightly modified – two terms are used e-cigarettes and electronic cigarettes (please use one consequently) Please use the term electronic cigarette (e-cigarette) when it appears the first time in the text and then consequently use the term e-cigarette The sentence in line 46-49 is very long – please create two sentences The aim of the study (line 51-54) is in the middle of Introduction and repeated in the last sentence of Introduction (line 84-86) The subheading “Study context” is not needed It is mentioned that1500 participants were recruited through an online commercial panel for marketing research and then wave 1, n=1501? The study sample (number of study participants) should be clearly described (in table 1 - 1073 participants are indicated but they are not clearly described/mentioned in section 2.1) In the section 2.2.1. all the questions are focusing on e-cigarettes whereas in the aim of the study NVPs are pointed out. Some more information can be added into the section 3. Statistical analyses (which variables were included in multivariate logistic regression? – this information is added under the table 2 but it also needs to be mentioned in the text) Tables 2 and 3 need to be corrected (AOR moved to 95%CI column) In the relevant tables - please change “Unadjusted odds ratio and Adjusted odds ratio” into “Unadjusted model and Adjusted model” and “Main effect” into “OR”Author Response
Thank you very much to the Reviewer for your very thoughtful comments, which allow us to improve our manuscript. We have addressed all your concerns, as is detailed below. Comment: The title would be slightly modified – two terms are used e-cigarettes and electronic cigarettes (please use one consequently) Please use the term electronic cigarette (e-cigarette) when it appears the first time in the text and then consequently use the term e-cigarette |
Answer: Following this comment from the Reviewer, the title was changed, as was our use of the terms throughout the manuscript. |
Comment: The sentence in line 46-49 is very long – please create two sentences |
Answer: The sentences were rewritten in the introduction section, page 1, lines 42-45. |
Comment: The aim of the study (line 51-54) is in the middle of Introduction and repeated in the last sentence of Introduction (line 84-86) |
Answer: We have removed this double aim in the introduction, page 2, lines 51-54. |
Comment: The subheading “Study context” is not needed It is mentioned that1500 participants were recruited through an online commercial panel for marketing research and then wave 1, n=1501? |
Answer: We removed the subheading. Page 2, line 68. In addition, a sentence specifying that the estimated sample size in each wave was set in 1500 respondents, page 2, lines 92-93, |
Comment: The study sample (number of study participants) should be clearly described (in table 1 - 1073 participants are indicated but they are not clearly described/mentioned in section 2.1) In the section 2.2.1. All the questions are focusing on e-cigarettes whereas in the aim of the study NVPs are pointed out. Some more information can be added into the section 3. |
Answer: In response to the Reviewer comment, we included in section 2.1 the sample size used in this study (page 3, line 98) and in the section 3 (page 4, lines 150-151). |
Comment: Statistical analyses (which variables were included in multivariate logistic regression? – this information is added under the table 2 but it also needs to be mentioned in the text) |
Answer: In response to this comment from the Reviewer, variables included in the multivariate logistic regression models, were also described in the text (Page 4, lines 145-148). |
Comment: Tables 2 and 3 need to be corrected (AOR moved to 95%CI column) In the relevant tables - please change “Unadjusted odds ratio and Adjusted odds ratio” into “Unadjusted model and Adjusted model” and “Main effect” into “OR” |
Answer: Tables 2 through 5 were updated to respond to this concern |

Reviewer 3 Report
This manuscript describes the prevalence and correlates of adult smokers’ discussions about nicotine vaping products (NVPs) with health professionals (HPs), including whether these discussions may lead smokers and vapers to use NVPs for smoking cessation. The study is simplistic and straightforward. Potential data biases and limitations are discussed.
Author Response
Comment: This manuscript describes the prevalence and correlates of adult smokers’ discussions about nicotine vaping products (NVPs) with health professionals (HPs), including whether these discussions may lead smokers and vapers to use NVPs for smoking cessation. The study is simplistic and straightforward. Potential data biases and limitations are discussed. |
Answer: Thank you very much to the Reviewer for the sincere and encouraging comment. |

Round 2
Reviewer 2 Report
I have no further comments